# Trends in the estimated proportion of outpatients with menstrual disorders and the number of prescribed low-dose estrogen/progestin drugs in Japan: A descriptive study

Motoyuki Nakao[1]*, Kotaro Kuwaki[1], Keiko Yamauchi[1], Kyoko Nomura[2],
Shinichi Tanihara[1]

1 Department of Public Health, School of Medicine, Kurume University, Kurume, Fukuoka, Japan,
2 Department of Environmental Health Science and Public Health, Akita University Graduate School of Medicine, Akita, Japan

* nakao_motoyuki@med.kurume-u.ac.jp

## Abstract

### Objectives

The aim of this study was to utilize publicly available government statistics to evaluate the trends in the estimated proportion of patients with menstruation-related diseases, including dysmenorrhea and endometriosis, and the number of low-dose estrogen/progestin drugs prescribed in Japan over approximately 20 years.

### Methods

The estimated proportion of outpatients with menstrual disorders, proportion of persons reporting irregular menstruation or menstrual pain, and number of prescriptions of low-dose estrogen/progestin drugs were extracted from the Patient Survey (1999–2023), Comprehensive Survey of Living Conditions (1998–2022), and National Database (NDB) Open Data (2014–2023), respectively.

### Results

According to the Patient Survey, the estimated proportion of outpatients with menstrual disorders, including dysmenorrhea, remained stable until 2011 and rose sharply after 2014. NDB data have demonstrated that the number of prescribed low-dose estrogen/progestin drugs covered by health insurance has increased since 2014. In contrast, the Comprehensive Survey of Living Conditions revealed that the rate of self-reported irregular menstruation or menstrual pain has continued to decline slightly over the past 20 years.

**Data availability statement:** All data underlying the findings described in our paper are available from figshare (DOI: 10.6084/m9.figshare.29148170). The original raw datasets analyzed during the current study are available from e-Stat (https://www.e-stat.go.jp/en) for the Patient Survey and the Comprehensive Survey of Living Conditions, and from the Ministry of Health, Labour and Welfare (https://www.mhlw.go.jp/stf/seisakunit-suite/bunya/0000177182.html) for the NDB Open Data. Direct links are available from the Supporting Information file.

**Funding:** This research was supported by Japan Society for the Promotion of Science (JSPS: https://www.jsps.go.jp/english/) Grant-in-Aid for Scientific Research (C) (Grant Number 23K01927 to MN) and Ministry of Health, Labour and Welfare of Japan (MHLW: https://www.mhlw.go.jp/stf/english/index.html) Health Labor Science Research Grant (Grant Number 23FB1002 to KN). There was no additional external funding received for this study. The funders had no role in study design, data collection and analysis, decision to publish, or preparation of the manuscript.

**Competing interests:** The authors have declared that no competing interests exist.

**Abbreviations:** LEP, low-dose estrogen/progestins; OTC, over-the-counter; NSAIDs, non-steroidal anti-inflammatory drugs; NDB, National Database; ICD, International Classification of Diseases; WHO, World Health Organization; NET, norethisterone; EE, ethynyl-estradiol; DRSP, drospirenone; LD, low-dose; ULD, ultra-low-dose; LNG, levonorgestrel; QoL, Quality of Life.

## Conclusions

The increased proportion of outpatients receiving treatment for menstrual disorders since 2014 coincides with the expansion of treatment options due to insurance coverage of low-dose estrogen/progestin drugs for dysmenorrhea. In addition, the launch of generic low-dose estrogen/progestin drugs since 2015 and the reduced economic burden due to lower drug prices may have led to an increased number of patients who continued treatment, resulting in an increased rate of follow-up visits. These findings provide a descriptive overview of national trends in menstrual health and healthcare utilization, but should be interpreted with caution given the limitations of aggregate data and the absence of individual-level or inferential statistical analysis.

## Introduction

Menstruation is a physiological phenomenon characterized by periodic bleeding through the vagina in reproductively mature women. It is an important health indicator for women, but many women experience abdominal pain and other symptoms of discomfort before and during menstruation [1–4].

Historically, women's health, particularly in relation to menstruation, has been both neglected and stigmatized, leading to embarrassment and marginalization. Furthermore, until a few decades ago, menstruation was less frequent than it is today, and fewer women experienced menstrual disorders. Consequently, women with menstrual health concerns have remained largely invisible. These factors have contributed to the lack of extensive research on menstrual health until recent years [1,5,6]. However, the number of women with menstruation-associated symptoms is currently higher than in the past [1,7,8], presumably due to an increase in the lifetime number of menstrual cycles. In recent years, a global consensus that menstruation is important for improving many social aspects necessary for a healthy society, including physical and mental health and gender equality, has been increasingly established [9].Menstruation-associated symptoms have recently received renewed attention, as they have been reported to not only lower the quality of life and cause serious illnesses but also incur economic losses due to decreased labor productivity [4,5,10–12]. Dysmenorrhea is the primary condition responsible for menstruation-associated symptoms. When symptoms such as abdominal cramps and lower back pain occurs during or around menstrual period are so severe that they interfere with daily life, the condition is medically defined as dysmenorrhea. Dysmenorrhea can be classified into primary dysmenorrhea, which is not caused by a gynecological disease, and secondary dysmenorrhea, which is caused by a gynecological disease such as endometriosis [7,8].

In Japan, birth rate has been declining over the past several decades. The total fertility rate declined from 3.65 in 1950 to 1.26 in 2022 in Japan [10]. Women who do not become pregnant or breastfeed experience more menstrual periods than those who do, which may be a reason for the increased incidence of endometriosis which causes dysmenorrhea [11]. In addition, the decreased prevalence of endometriosis

among women who have given birth suggests that the increased number of menstrual cycles among women who have not given birth may have increased the severity of menstruation-associated symptoms over the past decades [13]. In Japan, the treatment for dysmenorrhea at medical facilities mainly involves the prescription of low-dose estrogen/progestin combination (LEP) drugs [14]. LEP has been covered by public health insurance in Japan for the treatment of dysmenorrhea and endometriosis since 2010. However, most women with menstruation-associated symptoms self-manage and few visit a physician [15,16]. Those who do not see a doctor have stated that they do nothing in particular or take over-the-counter (OTC) medicines to cope with their symptoms [17,18]. A study comparing the effectiveness of LEPs prescribed by physicians to women with that of OTC nonsteroidal anti-inflammatory drugs (NSAIDs) purchased by women on their own reported a reduction in the impact of menstruation-associated symptoms on daily life only in women who were prescribed LEPs [17]. Thus, in women with menstruation-associated symptoms, a visit to a healthcare provider may improve menstruation-related absenteeism and presenteeism. However, no studies have provided reliable statistical information on the extent to which women with menstrual symptoms visit their doctors and are prescribed LEPs. The objectives of this descriptive study are twofold: first, to compare trends in the estimated proportion of outpatients with menstrual disorders (using medical facility-based data) and the proportion of persons reporting irregular menstruation or menstrual pain (using population-based self-report data); and second, to describe changes in LEP prescription patterns and their temporal relationship with outpatient visits for menstrual disorders. By integrating multiple national datasets, we aim to provide a comprehensive overview of menstrual health trends in Japan from both clinical and population perspectives.

## Methods

### Study design

This study is descriptive study for describing trend of the estimated number of patients with menstrual disorders and endometriosis, and prescription of LEPs using the Patient Survey, Comprehensive Survey of Living Conditions, and National Database (NDB) Open Data. All these surveys represent government statistics in Japan and are publicly available. Direct links to the original public datasets used in this study are provided in S1 File.

### Patient survey

The Patient Survey is conducted once every three years in Japan under the Statistics Act to clarify the actual conditions of patients who use medical facilities such as hospitals and clinics and to obtain basic data for medical administration. The subjects were patients who used the medical facility selected by random stratified sampling except for the hospitals with more than 500 bed (such hospitals are subject to an exhaustive survey) [19]. While the subjects of the survey are the patients who used the medical facilities, the responses to the survey are made by the administrators of each medical facility based on their medical records. Survey items include inpatient and outpatient statuses, treatment details, payment methods, and so on [20]. The estimated proportion of patients is defined as the estimated number of patients per day per 100,000 population and is used as an indicator of the number of patients. Patients are defined as individuals treated in hospitals, clinics, and dental clinics on the day of the survey. Data from the Patient Surveys in 1999, 2002, 2005, 2008, 2011, 2014, 2017, 2020 and 2023 were used in this study. The classification of injuries and diseases in the patient survey is based on the International Classification of Diseases (ICD-10) by the World Health Organization (WHO) [21].

The subclass menstrual disorders includes the following disorders with ICD-10 codes N91: absent, scanty, and rare menstruation (N91.0: primary amenorrhea; N91.1: secondary amenorrhea; N91.2: amenorrhea, unspecified; N91.3: primary oligomenorrhoea; N91.4: secondary oligomenorrhoea; N91.5: oligomenorrhoea, unspecified); N92: hypermenorrhea, frequent menstruation, irregular menstruation excluding postmenopausal bleeding (N95.0) (N92.0: excessive and frequent menstruation with regular cycle; N92.1: excessive and frequent menstruation with irregular cycle; N92.2: excessive menstruation at puberty; N92.3: ovulation bleeding; N92.4: excessive bleeding in the premenopausal period; N92.6: irregular

menstruation, unspecified); N94: pain and other conditions associated with female genital organs and menstrual cycle (N94.0: Mittelschmerz; N94.3: premenstrual tension syndrome; N94.4: primary dysmenorrhea; N94.5: secondary dysmenorrhea; N94.6: dysmenorrhea, unspecified; N94.8: other specified conditions associated with female genital organs and menstrual cycle; N94.9: unspecified condition associated with female genital organs and menstrual cycle). The subclass endometriosis includes the following conditions with ICD-10 code N80: endometriosis (N80.0: endometriosis of uterus; N80.1: endometriosis of ovary; N80.2: endometriosis of fallopian tube; N80.3: endometriosis of pelvic peritoneum; N80.5: endometriosis of intestine; N80.8: other endometriosis; N80.9: endometriosis, unspecified). The estimated proportion of patients was extracted for the sub-categories menstrual disorders and endometriosis in the injury and disease sub-categories of the Patient Survey.

## Comprehensive survey of living conditions

The Comprehensive Survey of Living Conditions is conducted in Japan under the Statistics Act to obtain the basic data necessary for planning and administering the nation's health, medical care, welfare, pensions, income, and other basic matters of living conditions through the Health, Labor and Welfare Administration. It is conducted annually, with a larger survey conducted once every three years. The subjects for the Health Questionnaire Survey were households and their members, selected through stratified random sampling. For the 2022 survey, for example, approximately 300,000 households were selected from a total of 54.3 million households in Japan. Of these, 205,063 households responded. Ultimately, data from 203,819 households were included in the final analysis, excluding those deemed ineligible for tabulation [22]. Health-related surveys were included as the survey items only in large-scale surveys. The health questionnaire included information on subjective symptoms, hospital visits, health consciousness, mental health, and cancer-screening status [22]. In this study, the proportion of persons who reported any symptom was used as an indicator of the number of persons with self-reported symptoms. The estimated proportion of people who reported symptoms was calculated as the number of people who reported symptoms per 1,000 population. The number of persons who reported symptoms was the number of persons in the population, excluding hospitalized persons who were aware of their illnesses or injuries. This study used data from the Comprehensive Survey of Living Conditions for 1998, 2001, 2004, 2007, 2010, 2013, 2016, 2019, and 2022. From this survey, the proportion of persons who reported symptoms of irregular menstruation or menstrual pain per 1,000 persons was extracted. Respondents selected more than one applicable symptom for this question from a list of 41 symptoms and "others." For the 2001 survey, data on the proportion of persons who reported irregular menstruation or menstrual pain, classified according to sex and age, were not available online.

## NDB open data

NDB is an abbreviation for the National Database of Health Insurance Claims and Specific Health Checkups of Japan, a highly complete database that includes health insurance claims information and the results of specified health examinations and specific health guidance collected from health insurers in Japan. The NDB Open Data refer to a tabulation of aggregated basic information extracted from the NDB that does not contain confidential information and is maintained by the Ministry of Health, Labour and Welfare [23]. The NDB Open Data include tables outlining medical practices and their medical remuneration points, dental disease results from specific health checkups, and drug prescriptions. Data on drugs were disclosed for outpatient and inpatient oral, topical, and injectable drugs, including the top 100 drugs with the highest prescription quantities for each three-digit drug classification based on the unit of the national health insurance drug price standard listing. This study used the number of prescriptions (2014–2023) of LEPs used only in the treatment of dysmenorrhea and endometriosis under insurance coverage as an indicator. The LEPs utilized in this study included norethisterone/ethinylestradiol (NET/EE), drospirenone/ethinylestradiol (DRSP/EE), and levonorgestrel/ethinylestradiol (LNG/EE) combinations. All of these LEPs are administered orally.

## Statistical analysis

To account for changes in the age distribution of the female population over time, age-standardized rates were calculated for both the estimated proportion of outpatients with menstrual disorders and endometriosis (from the Patient Survey) and the proportion of persons reporting irregular menstruation or menstrual pain (from the Comprehensive Survey of Living Conditions). Age-specific rates were calculated for each 5-year age group, and the 2015 model female population in Japan was used as the standard population. The age-standardized rate for each survey year was obtained by weighting the age-specific rates by the corresponding age group proportions in the standard population and summing these weighted rates [24].

## Ethical considerations

All surveys comprised government statistics publicly available on the Internet, all data were aggregated, and no personal information was included. Therefore, this study is not subject to the current ethical guidelines and did not require an ethical review.

## Results

### Trends in the estimated proportion of outpatients with menstrual disorder and endometriosis

Trends in the rate of estimated outpatients for the subcategories, menstrual disorder and endometriosis, were examined using data from the Patient Survey (Fig 1). The estimated proportion of outpatients with all injuries and diseases increased slightly during the first visit, while that for follow-up visits remained unchanged with some fluctuations (Fig 1A, dotted line). For menstrual disorder, the estimated proportion of outpatients visiting for follow-up remained almost unchanged until 2011, whereas the estimated proportion of outpatients in the 2014 survey was approximately double that in 2011 and has continued to increase since then (Fig 1B, dotted line). Finally, the estimated proportion of outpatients visiting for follow-up in 2023 was 32 per 100,000 females, which was 5.3 times the proportion in 1999. For endometriosis, the estimated proportion of outpatients on their first visit remained almost unchanged over all surveys, and for follow-up has continued to increase slightly since 2005 (Fig 1C, dotted line). After adjusting for changes in the age distribution of the female population, the age-standardized trends in the estimated proportion of outpatients with menstrual disorders and endometriosis were similar to the crude trends (see solid lines in Fig 1A–C). The increase in follow-up visits for menstrual disorders and endometriosis since 2014 remained evident after age standardization. The estimated proportion of outpatients peaked in the younger and older age groups for first-time outpatients with all injuries and diseases, with broad peaks for patients in their middle ages (Fig 1D). For menstrual disorders, the age distribution of the outpatients on the first visit was peaked among those in their 20s in most cases (Fig 1E). The age distribution of the outpatients on the first visit for endometriosis varied widely depending on the year of the survey (Fig 1F). Although the estimated proportion of outpatients who visited for follow-up for all injuries and diseases was higher in the older age groups (Fig 1G). The estimated proportion of outpatients visiting for follow-up due to menstrual disorders often peaked among those in their late 20s (Fig 1H). The estimated proportion of outpatients visiting for follow-up due to endometriosis was higher among those in their 40s (Fig 1I).

### Trends in the proportion of persons who reported irregular menstruation or menstrual pain

According to the Comprehensive Survey of Living Conditions, the proportion of persons who reported any symptom showed a slight upward trend until 2007, after which the proportion began to decline (Fig 2A). The proportion of persons who reported symptoms of irregular menstruation or menstrual pain was highest in 2007, and then decreased until the 2013 survey, after which the proportion remained flat or slightly decreased (Fig 2B). The age distribution of persons who reported any symptoms decreased annually but indicated a higher proportion of persons in the older age groups (Fig 2C).

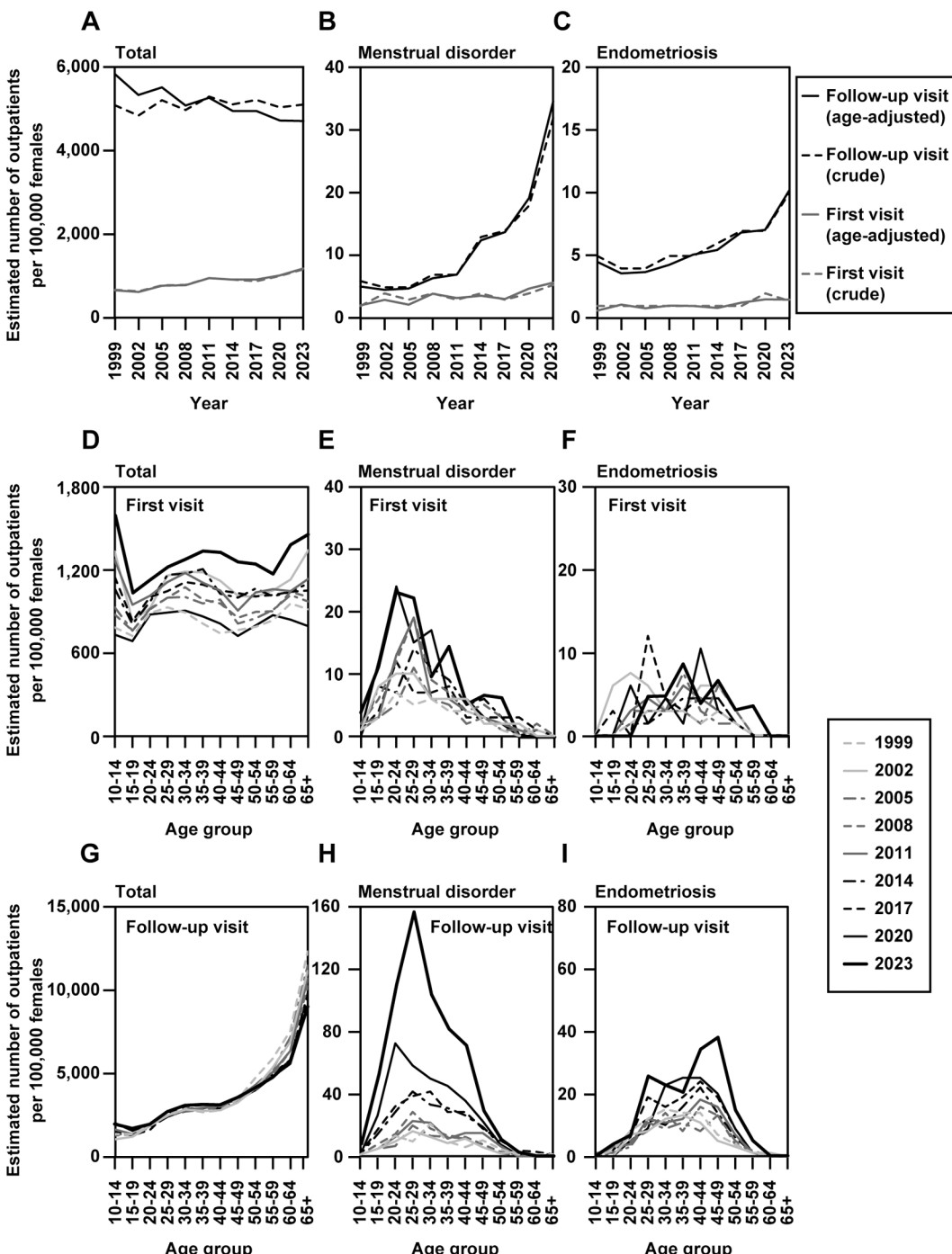

**Fig 1. Trends in the estimated proportion of outpatients from the Patient Survey (1999–2023).** Trends in the estimated proportion of outpatients with symptoms including all injuries and diseases (A), menstrual disorder (B), and endometriosis (C) among women. Trends in the age distribution of outpatients on their first visit for all injuries and diseases (D), menstrual disorder (E), and endometriosis (F) among women and on their follow-up visit for all injuries and diseases (G), menstrual disorder (H), and endometriosis (I) among women.Note: In panels A–C, solid lines represent age-adjusted rates, and dotted lines represent crude rates. In panels D–I, lines corresponding to the year 2023 are shown as bold solid lines to highlight the most recent data.

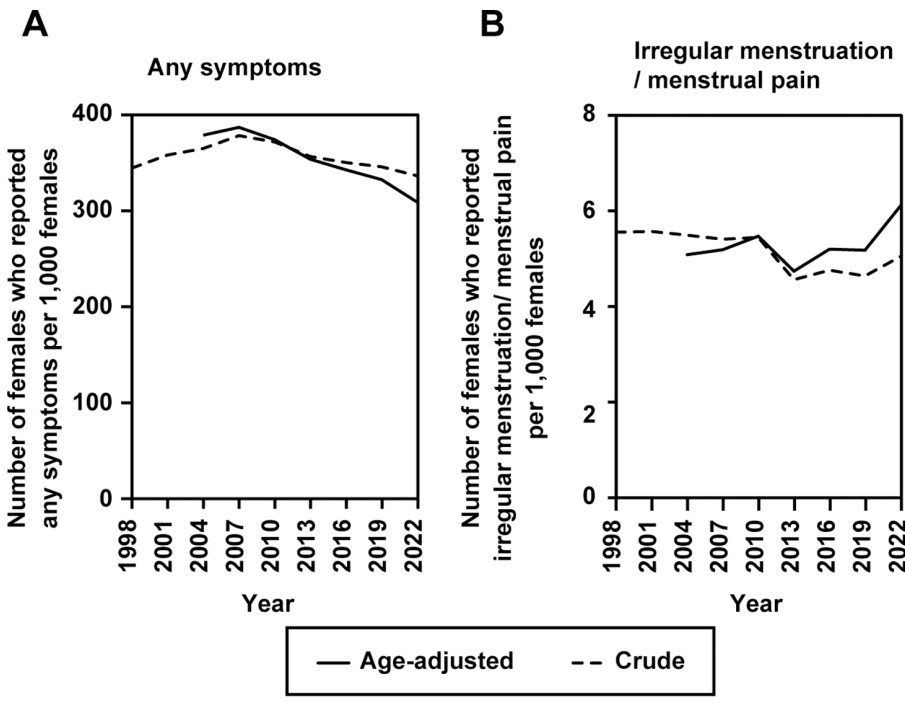

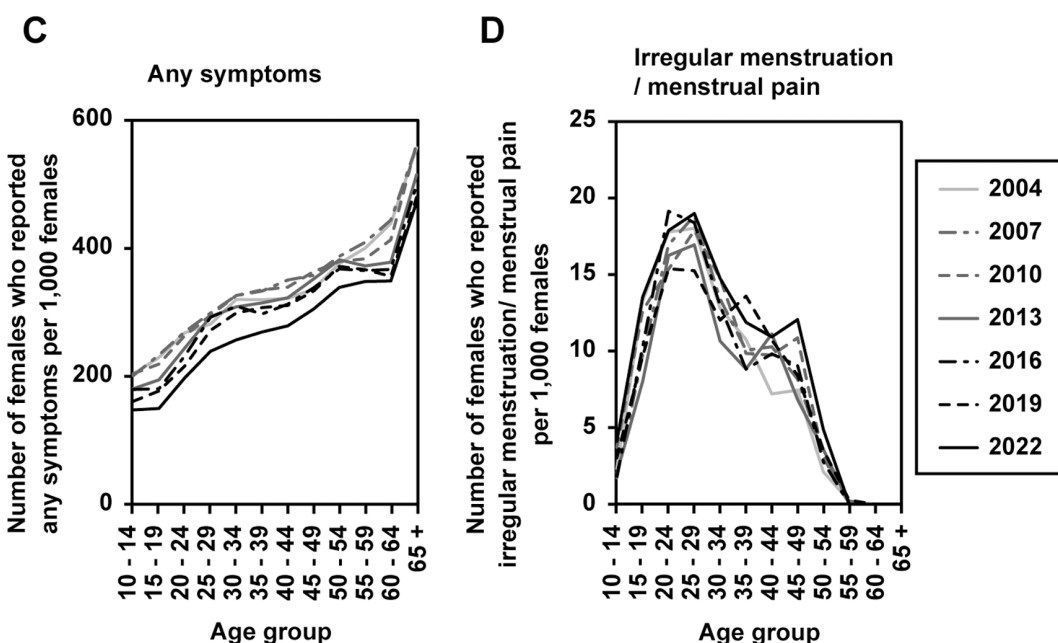

**Fig 2. Trends in the proportion of persons who reported symptoms from the Comprehensive Survey of Living Conditions (1998–2022).** Trends in the proportion of persons who reported any symptoms (A) and irregular menstruation or menstrual pain (B). Trends in the age distribution of persons who reported any symptoms (C) and irregular menstruation or menstrual pain (D). * The 2001 survey lacked data on the proportion of individuals who reported irregular menstruation or menstrual pain.

A bimodal age distribution was observed among persons who reported menstrual irregularities and menstrual pain, with peaks among persons in their early 20s and late 40s (Fig 2D).

**Trends in the number of prescriptions and drug prices for LEPs**

According to the NDB Open Data, the number of LEP prescriptions increased 5-fold from 2014 to 2023 (Fig 3A). In 2014, NET/EE (35 μg) was the most prescribed LEP, whereas from 2015 to the present, DRSP/EE is the most prescribed. NET/EE was predominantly prescribed at low doses (LD) with an estrogen content of 35 μg from 2014 to 2018, while prescriptions at very low doses (ULD) with an estrogen content of 20 μg became more common from 2019. The prescription volume of the LNG/EE also increased when treatments for dysmenorrhea and endometriosis began to be covered by health insurance from 2019 onwards. All LEPs launched to date, regardless of the type of drug formulation, have increased in number. Regarding age distribution, LEPs were most frequently prescribed to patients in their 20s from 2014, with the most apparent increase seen in 2023 (Fig 3B). By 2023, the generic low-dose estrogen/progestin (LEP) formulations available in Japan included norethisterone/ethinylestradiol (NET/EE) at both low-dose (LD) and ultra-low-dose (ULD) levels, with the LD and ULD generics launched in 2015 and 2018, respectively, as well as drospirenone/ethinylestradiol (DRSP/EE), for which the generic product became available in 2022. After the introduction of these generics, prescriptions for the original brand-name products of both NET/EE (LD and ULD) and DRSP/EE were progressively replaced by their generic counterparts, resulting in a marked decline in the number of prescriptions for brand-name drugs (Fig 3C).The prices of both original and generic drugs have been declining over the years, with generic drugs priced at approximately half the price of the original drugs (Fig 3D).

## Discussion

In this study, we used data from publicly available sources to examine the estimated proportion of patients with menstrual disorders, proportion of persons who reported menstruation-associated symptoms, and number of prescriptions for LEPs used to treat dysmenorrhea and endometriosis. According to the Patient Survey, the estimated proportion of outpatients followed up for menstrual disorders and endometriosis has increased over the years (Figs 1A and 1B), with the increase being particularly noteworthy for menstrual disorders since the 2014 survey among young adults in their 20s. During the study period (1999–2023), the overall estimated proportion of patients with ailments including all injuries and diseases remained almost unchanged, indicating that reports of menstrual disorders increased specifically and independently of the estimated proportion of patients with all injuries and diseases. On the other hand, the Comprehensive Survey of Living Conditions shows that the proportion of persons who reported any symptom (1998–2022) decreased slightly in terms of both those who reported any symptoms and those who reported menstrual irregularities and menstrual pain (Fig 2A). Thus, the data suggest that the number of people who are aware of menstruation-associated symptoms has not increased over the past twenty or more years. This inference is supported by general lack of change in the estimated proportion of outpatients visiting for first-time menstrual disorders over the past 20 years, with some fluctuations (Fig 1B). The estimated proportion of outpatients following up for menstrual disorders has risen sharply since the 2014 survey, indicating that the number of continuous visits due to menstrual disorders has increased significantly in recent years. Importantly, age-standardized analyses showed that these trends remained evident even after adjusting for changes in the age distribution of the female population (see Fig 1A–C). This suggests that the recent increase in follow-up visits for menstrual disorders and endometriosis cannot be explained solely by demographic shifts, but likely reflects other factors such as changes in health-seeking behavior, insurance coverage, or public awareness.

The divergence between the increasing rate of medical follow-ups and the stagnation or slight decline in self-reported symptoms may reflect changes in health-seeking behavior, increased public awareness, reduced stigma, and improved access to care. [15–18,25,26]The increase in the number of follow-up visits for menstrual disorders can be attributed to the fact that LEPs were covered by public health insurance for dysmenorrhea since 2010 in Japan. According to the

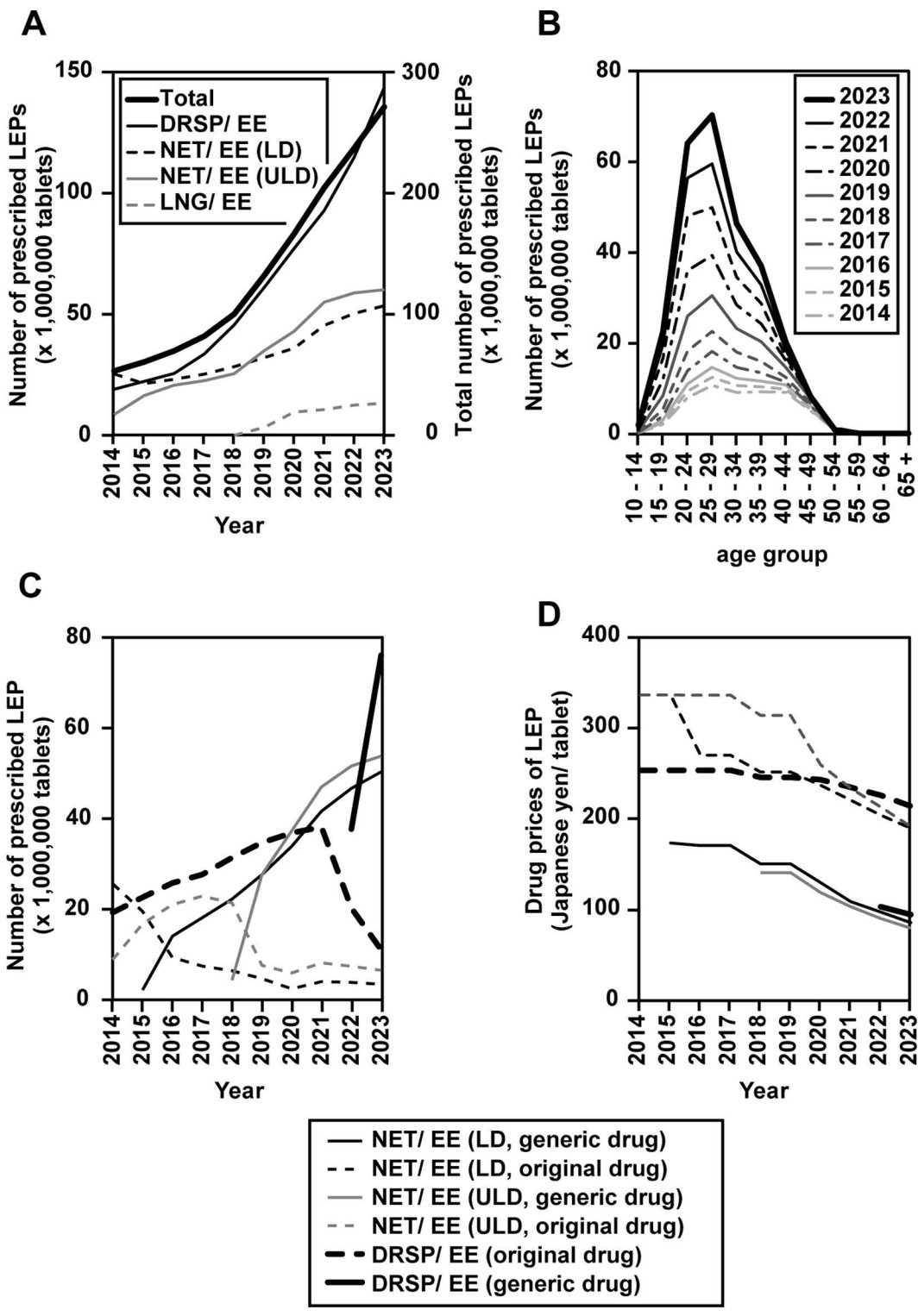

**Fig 3. Trends in the number of LEP prescriptions according to the NDB Open Data and the drug price of NET/EE and DRSP/EE (2014–2023).** Trends in the number of LEP prescriptions by constituent (A). Trends in the number of prescriptions for LEPs by age distribution (B). Trends in the number of original and generic LD and ULD NET/EE drug prescriptions (C). Trends in the price of LD and ULD NET/EE and DRSP/EE drugs (D). Note: In all panels, the lines representing data from 2023 are shown as bold solid lines to emphasize the most recent trends.

NDB Open Data, prescriptions for LEPs have risen sharply since 2014, with a particularly large increase among patients in their 20s (Figs 3A and 3B), similar to the increase in the estimated proportion of outpatients with menstrual disorders (Figs 1A and 1C). Currently, LEP prescriptions in Japan are covered by the public health insurance only for endometriosis and dysmenorrhea. Thus, the increase in the estimated proportion of outpatients with menstrual disorders is associated with an increase in LEP prescriptions since 2014 in particular. Current options for the treatment of dysmenorrhea include LEPs, NSAIDs, and traditional Chinese medicines [14]. LEPs are prescription drugs that effectively treat dysmenorrhea [17,27–29]. In contrast, NSAIDs and traditional Chinese medicines are available as OTC drugs, are not expensive, and can be easily administered on an *ad hoc* basis when menstrual symptoms appear. However, OTC drugs are ineffective in alleviating the effects of menstruation-associated symptoms in daily life [17]. This may be one reason for the low estimated proportion of outpatients who visited their healthcare providers for follow-up visits with menstrual disorders before LEPs were covered by public health insurance for dysmenorrhea. In other words, the patient may not have been willing to continue visiting a physician to receive a prescription for a drug that is equivalent to an OTC drug without any perceptibly significant benefit. Despite the availability of effective prescription treatments, many women continue to rely on OTC medications for menstrual pain. This preference may be attributed to the convenience and accessibility of OTC drugs, as well as practical or economic barriers to clinic visits [15–18]. Cultural stigma and societal norms in Japan also contribute to the low rate of healthcare-seeking for menstrual disorders. Menstruation is often considered a private or taboo topic, and many women—especially adolescents and young adults—feel embarrassed to discuss menstrual symptoms or to visit a gynecologist for these issues. The expectation to endure menstrual pain as a normal part of womanhood, combined with limited reproductive health education and the perception that gynecology clinics are primarily for pregnancy or older women, further discourages medical consultation. These cultural and societal barriers have been well documented and are thought to play a significant role in healthcare avoidance for menstrual symptoms in Japan [15–18,25,26].In the Patient Survey, patients with dysmenorrhea classified as menstrual disorder may include a substantial number of patients with secondary dysmenorrhea who potentially have conditions in the reproductive organs. For example, morphological abnormalities of the reproductive organs have been reported to be closely associated with the development of endometriosis [30]. Endometriosis is often difficult to diagnose in young women, even when symptoms such as menstrual cramps are present, and diagnosis can frequently take several years. Given the risks of infertility and decreased quality of life (QoL) for patients, early differential diagnosis of endometriosis is crucial [31,32]. From this perspective, the increasing number of women who continue to seek medical care is beneficial for those suffering from menstruation-associated symptoms.

Furthermore, generic drug options for LEPs have been increasing recently, and generic LEP prescriptions are rapidly replacing the original drugs (Fig 3C). The increased estimated proportion of outpatients following up their menstrual disorders may also be due, in part, to the contribution of generic drugs. Because LEPs are consumed continuously for many days, the economic burden of drug costs can be non-negligible for women with menstruation-associated symptoms. Since the price of generic drugs for LEPs is approximately half the price of original drugs (Fig 3d), the widespread use of generic drugs may contribute to the reduction of women's economic burdens. In fact, according to a report by the Health and Global Policy Institute in Japan, approximately 10% of women who had menstruation-associated symptoms did not visit an obstetrician/gynecologist because they felt that the cost of treatment was too high [18]. Additionally, since patients with primary dysmenorrhea tend to be relatively young women [13], and the burden of continuous medical expenses is considerable for students and young women who have recently left their families. Therefore, the reduction of the financial burden due to the launch of generic drugs may be the reason why follow-up visits are increasingly being conducted at a younger age.

In addition to the expansion of insurance coverage for LEPs, several other factors may have contributed to the observed increase in follow-up visits for menstrual disorders. Recent studies have highlighted the importance of social and cultural factors, such as increased public awareness, reduced stigma surrounding menstruation, and evolving attitudes toward women's health, in influencing healthcare-seeking behavior among women [25,33,34]. In Japan, healthcare avoidance for menstrual symptoms has been associated with attitudes of endurance, limited health literacy, and lack

of awareness about available treatment options [25]. Improvements in diagnostic capabilities and greater emphasis on menstrual health in public discourse may also have encouraged more women to seek medical care, even for symptoms previously considered normal. These findings are consistent with international literature showing that social context, education, and economic factors can significantly affect access to and utilization of menstrual health services [33,34]. Therefore, while the increased use of LEPs and insurance coverage are important, these broader social and systemic changes should also be considered when interpreting recent trends.

In Japan, LEPs are currently covered by public health insurance only for dysmenorrhea and endometriosis [14]. However, LEPs have demonstrated efficacy for other menstrual-related conditions, such as abnormal uterine bleeding, premenstrual syndrome, and polycystic ovary syndrome [35–37]. If insurance coverage were expanded to include these indications, it could improve access to effective treatment for a wider range of women and potentially reduce the burden of untreated menstrual symptoms. Such a policy change may lead to improvements in quality of life and productivity, but would also require careful consideration of healthcare costs and the development of appropriate clinical guidelines to ensure rational use.

The national survey data used in this study provide a foundation for monitoring and predicting future trends in menstrual health in Japan. By tracking changes in outpatient visits, self-reported symptoms, and prescription patterns over time, it is possible to identify emerging needs and gaps in care, particularly among younger women. The observed increase in LEP prescriptions and follow-up visits among women in their 20s suggests improved access, but the continued reliance on OTC medications and the stagnation in self-reported symptoms indicate that barriers such as cost, stigma, and lack of health literacy persist. These findings highlight the importance of ongoing surveillance and the need for targeted interventions—such as health education, outreach, and policy reforms—to further improve healthcare accessibility and menstrual health outcomes for younger women. Future studies could also leverage digital health data and predictive modeling to better forecast demand and inform resource allocation [38,39].

This study has several limitations. First, this study was conducted using only publicly available data. The Patient Survey, Comprehensive Survey of Living Conditions, and NDB Open Data present only aggregated data and thus cannot be analyzed in detail. For example, with regard to the classification of injuries and diseases, the Patient Survey report cannot distinguish between dysmenorrhea and amenorrhea, which are included in the same classification as menstrual disorders. An additional concern is that, in patient surveys, the inability to differentiate between primary and secondary dysmenorrhea may result in dysmenorrhea caused by endometriosis being classified as menstrual disorders rather than as endometriosis.

Second, the data used in this study differed from each statistical survey in terms of participants, methods, and the year of survey. Patient surveys are reported by medical facilities and are, therefore, more reliable for diagnosing injuries and illnesses. On the other hand, the proportion of persons who reported symptoms in the Comprehensive Survey of Living Conditions is based on subjective symptoms reported by the persons themselves and not on diagnoses made by physicians; therefore, the reliability of the answers is weak, and it is unclear what diseases exist behind the symptoms reported as menstrual irregularities and menstrual pain.

Third, the statistical surveys used in this study were conducted at various time points. The Patient Survey is conducted once every three years when medical facilities report the records of those who visited a medical facility on a particular day. The Comprehensive Survey of Living Conditions is a survey of randomly selected household members, and a health questionnaire is administered every three years during a large-scale survey. These surveys were conducted in different years. Therefore, it was not possible to compare the survey results for each year. Although these surveys were conducted over time, their panels varied annually; therefore, they did not follow the progress of the same person. The NDB Open Data is the basic aggregate of NDB data released to the public once a year. In this study, we used approximately 20 years of data from the Patient Survey and Comprehensive Survey of Living Conditions, but data older than these were not available because the NDB Open Data was available only since the fiscal year 2014. Therefore, the number of LEP

prescriptions prior to this year remains unknown. Nevertheless, the use of multiple national datasets in this study was intended to provide a comprehensive, descriptive overview of menstrual health trends in Japan from both clinical (medical facility-based) and population (self-reported) perspectives. Because the data sources, definitions, and units differ, no statistical correlation or causal inference was attempted between the prevalence of menstrual disorders and LEP prescriptions. Rather, we sought to highlight temporal trends and potential gaps between self-reported symptoms and medical care utilization. This descriptive approach allows for the identification of emerging needs and changes in health-seeking behavior, but the observed associations should be interpreted with caution. Although these limitations could have been overcome by conducting a more detailed study using more complete data, such as health insurance claims data, this study was conducted as a pilot study, giving priority to simplicity and using only publicly available data to examine trends in the indicators of menstruation-related diseases. Furthermore, because our analyses were based on aggregated data from government statistics, we were unable to assess individual-level treatment adherence, long-term persistence, symptom improvement, or patient-reported outcomes such as quality of life. Therefore, while we observed a substantial increase in LEP prescriptions and a reduction in economic burden due to generic drugs, we cannot directly determine whether these changes translated into improved clinical outcomes or long-term adherence. Future studies using individual-level data, such as health insurance claims or electronic medical records, are needed to clarify the real-world effectiveness of LEP treatment for menstrual disorders and the impact of generic drug introduction on adherence.

Finally, because the aggregated data used in this study do not allow for the distinction between primary and secondary dysmenorrhea, it is possible that some cases of secondary dysmenorrhea, such as undiagnosed endometriosis or adenomyosis, may remain included in the "menstrual disorders" category, while diagnosed cases are reclassified under "endometriosis" or other relevant categories. Therefore, a certain proportion of secondary dysmenorrhea may be excluded from the follow-up group, but complete differentiation is not possible. Future research using individual-level data is needed to address this issue.

Although these limitations could have been overcome by conducting a more detailed study using more complete data, such as health insurance claims data, this study was conducted as a pilot study, giving priority to simplicity and using only publicly available data to examine trends in the indicators of menstruation-related diseases.

## Conclusions

In conclusion, the estimated proportion of outpatients with menstrual disorder in the Patient Survey showed a remarkable increase in follow-up visits since the 2014 survey. This increase was attributable to LEPs being covered by public health insurance for treating dysmenorrhea in 2010, the subsequent increase in LEP options, and the reduction in the economic burden of LEPs owing to the launch of generic LEP drugs. The socially perceived importance of women's health, particularly menstruation-related health problems and menopause, is expected to increase in the coming years. This study is expected to contribute as a preliminary survey for more detailed studies in the near future and provide fundamental data for the formulation of health care policies regarding menstruation-related diseases.

## Supporting information

**S1 File. Direct links to original datasets.** This file contains a list of direct hyperlinks to the publicly available original datasets used for this study, including the Patient Survey, the Comprehensive Survey of Living Conditions, and the National Database (NDB) Open Data.
(DOCX)

## Acknowledgments

We would like to thank Editage (www.editage.jp) for English language editing.

## Author contributions

**Conceptualization:** Motoyuki Nakao.

**Data curation:** Motoyuki Nakao.

**Formal analysis:** Motoyuki Nakao.

**Funding acquisition:** Motoyuki Nakao, Kyoko Nomura, Shinichi Tanihara.

**Investigation:** Motoyuki Nakao, Kotaro Kuwaki.

**Methodology:** Motoyuki Nakao.

**Project administration:** Motoyuki Nakao.

**Supervision:** Kyoko Nomura, Shinichi Tanihara.

**Validation:** Motoyuki Nakao.

**Visualization:** Motoyuki Nakao.

**Writing – original draft:** Motoyuki Nakao.

**Writing – review & editing:** Motoyuki Nakao, Kotaro Kuwaki, Keiko Yamauchi, Kyoko Nomura, Shinichi Tanihara.

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
