## [Decision Letter · Decision Letter 0]

PONE-D-24-41343Trends in the estimated proportion of outpatients with menstrual disorders and the number of prescribed low-dose estrogen/progestin drugs in Japan: A descriptive studyPLOS ONE

Dear Dr. Nakao,

Thank you for submitting your manuscript to PLOS ONE. After careful consideration, we feel that it has merit but does not fully meet PLOS ONE’s publication criteria as it currently stands. Therefore, we invite you to submit a revised version of the manuscript that addresses the points raised during the review process.

We look forward to receiving your revised manuscript.

Kind regards,

Marcio Alexandre H. Rodrigues, Ph.D

Academic Editor

PLOS ONE

Journal Requirements:

“This research was supported by Japan Society for the Promotion of Science (JSPS: https://www.jsps.go.jp/english/) Grant-in-Aid for Scientific Research (C) (Grant Number 23K01927 to MN) and  Ministry of Health, Labour and Welfare of Japan (MHLW: https://www.mhlw.go.jp/stf/english/index.html) Health Labor Science Research Grant (Grant Number 23FB1002 to KN).”

“This research was supported by Japan Society for the Promotion of Science (JSPS: https://www.jsps.go.jp/english/) Grant-in-Aid for Scientific Research (C) (Grant Number 23K01927 to MN) and  Ministry of Health, Labour and Welfare of Japan (MHLW: https://www.mhlw.go.jp/stf/english/index.html) Health Labor Science Research Grant (Grant Number 23FB1002 to KN).”

4. Thank you for uploading your study's underlying data set. Unfortunately, the repository you have noted in your Data Availability statement does not qualify as an acceptable data repository according to PLOS's standards.

Additional Editor Comments:

The results presented in this study " Trends in the estimated proportion of outpatients with menstrual disorders and the number of prescribed low-dose estrogen/progestin drugs in Japan: A descriptive study" offer a comprehensive overview of national trends in outpatient visits for menstrual disorders and endometriosis, symptom reporting in population-based surveys, and the utilization and pricing of low-dose estrogen-progestin (LEP) contraceptives over the past two decades.

While the descriptive findings are valuable, several critical issues regarding internal consistency, methodological robustness, and interpretative depth require further consideration.

#1. One of the most prominent concerns arises from the divergence between trends in self-reported symptoms and outpatient visits. According to the Patient Survey data, the proportion of follow-up outpatient visits for menstrual disorders and endometriosis has markedly increased since 2014. In contrast, data from the Comprehensive Survey of Living Conditions indicate that the proportion of individuals reporting irregular menstruation or menstrual pain peaked in 2007 and subsequently declined or stabilized.This incongruity raises important questions:

• Does the increase in outpatient visits reflect improved diagnosis and management rather than a true rise in disease burden?

• Could changing diagnostic thresholds, increased public awareness, or health system reforms (e.g., insurance coverage for LEPs starting in 2019) have influenced patterns of care-seeking behavior?

• Are individuals with milder symptoms less likely to report them in surveys while still engaging with the healthcare system?

Without addressing these inconsistencies, the narrative on disease burden and healthcare utilization remains incomplete.

#2. The results indicate age-specific peaks in outpatient visits and symptom reporting, with notable shifts over time. For example, follow-up visits for menstrual disorders showed an unexpected increase among individuals in their early 20s in the 2020 survey, while endometriosis-related follow-ups increased among women in their 30s. However, the study does not offer a thorough explanation or control for external variables, such as demographic shifts or public health initiatives, which may have impacted these trends. Further stratified analyses or age-standardized rates would enhance interpretability. Could you discuss these questions?

#3. The observed 3.8-fold increase in LEP prescriptions from 2014 to 2021, particularly among younger women, is attributable to enhanced accessibility following expanded insurance coverage. Nonetheless, the results focus primarily on prescription volume and drug pricing, without addressing treatment adherence, efficacy, or patient-reported outcomes. The clinical relevance of increased LEP use—whether it translates to improved symptom control, reduced disease progression, or better quality of life—remains speculative without such data. What do you think about this point?

3.1. Moreover, while drug utilization data are detailed, the lack of correlation with outpatient visit or symptom trends weakens the integrated understanding of treatment effectiveness. Could you give us new insights on this point?

#4. The discussion and conclusion sections present a well-structured narrative linking the increased use of LEPs (low-dose estrogen–progestin combinations) to policy changes and access to affordable treatment. However, a deeper academic analysis raises important questions that were not sufficiently explored. First, while the association between LEP coverage and the rise in follow-up visits is clearly outlined, causality is assumed rather than critically interrogated. Were other potential contributing factors—such as increased public awareness, evolving cultural attitudes toward menstruation, or improvements in diagnostic capabilities—considered and ruled out? The study would benefit from discussing possible confounding variables that might influence health-seeking behaviors.

4.1. Moreover, the authors note a discrepancy between increasing medical follow-ups and the stagnation or slight decline in self-reported symptoms; however, this paradox is only superficially addressed. This inconsistency raises questions about the nature of symptom perception versus medical diagnosis. Could there be a growing recognition of menstrual disorders as legitimate medical conditions, even if subjective experiences remain stable? Alternatively, are there socioeconomic or educational factors influencing who seeks treatment and who does not? These gaps suggest that further qualitative research might help clarify why follow-up care has increased without a corresponding rise in symptom reporting.

#5. Minor revisions:

-How can future studies address the inability to distinguish between primary and secondary dysmenorrhea in aggregated data?

- What are the underlying reasons why many women still opt for over-the-counter (OTC) medications instead of seeking medical treatment for menstrual disorders?

- How significant is the reduction in economic burden due to generic drugs, and how does this affect long-term adherence to treatment?

- Are there cultural stigmas or societal norms in Japan that discourage women from visiting healthcare providers for menstrual-related symptoms?

- Could the insurance coverage for LEPs be extended to include other menstrual-related conditions, and what would be the potential impact?

- How can the data from this study be used to predict future trends in menstrual health and improve healthcare accessibility for younger women?

Reviewers' comments:

Reviewer's Responses to Questions

**Comments to the Author**

1. Is the manuscript technically sound, and do the data support the conclusions?

Reviewer #1: Partly

2. Has the statistical analysis been performed appropriately and rigorously? 

Reviewer #1: N/A

3. Have the authors made all data underlying the findings in their manuscript fully available?

Reviewer #1: No

4. Is the manuscript presented in an intelligible fashion and written in standard English?

Reviewer #1: Yes

5. Review Comments to the Author

Reviewer #1: An interesting work on the part of the authors. However, in my opinion, the study has two major problems:

1. The design of the study includes two research questions:

First question: Why the proportion of outpatients with menstrual disorders obtained from the Patient Survey (1999-2020) diverges from the proportion of persons reporting irregular menstruation or menstrual pain, obtained from the Comprehensive Survey of Living Conditions (1998-2022)? The answer to this question could be developed in a separate article. This would lead to the literature on self-perceived health status and the psychocultural conditioning factors that determine it. Moreover, to explore those conditioning factors from a women’s health perspective could be a promising research line.

Second question: Is the prevalence of menstrual disorders (regardless of the database used) and the prescription of LEPs obtained from NDB Open Data statistically correlated? Are there are clinical and epidemiological reasons to assume so? in my opinion the article should better justify the reason why this association is assumed.

If the research question is the first one, this implies a completely different article. Conversely, if the research question is the second one, I don’t see the reason why to use two different datasets to infer the prevalence of menstrual disorders.

2. The article claims that no studies have provided reliable statistical information on the extent to which women with menstrual symptoms visit their doctors and are prescribed LEPs. However, according to the authors the study is purely descriptive. I don’t see reliable statistics. In fact, the usage of publicly available aggregate data and the absence of a minimal statistical analysis reduces the level of evidence that supports the conclusions. I think the authors made a significant effort to investigate, aggregate and describe the different sources of data related to the menstrual disorders and that is the contribution of the article.

6. PLOS authors have the option to publish the peer review history of their article (what does this mean? ). If published, this will include your full peer review and any attached files.

**Do you want your identity to be public for this peer review?** For information about this choice, including consent withdrawal, please see our Privacy Policy .

Reviewer #1: No

---

## [Author Response · Author response to Decision Letter 1]

5 Jun 2025

Dear Dr. Marcio Alexandre H. Rodrigues, Academic Editor, PLOS ONE,

Thank you for your thoughtful review of our manuscript and for the opportunity to revise our work. We appreciate the time and expertise that you and the reviewer have dedicated to providing constructive feedback. We have carefully addressed all comments and suggestions, and we believe the manuscript has been substantially improved as a result.

Below, we provide a point-by-point response to all comments, detailing the changes made to the manuscript. Changes are highlighted in the revised manuscript with track changes.

UPDATE REGARDING NEW DATA:

While this manuscript was under review, the 2022 and 2023 NDB Open Data, and 2023 Patient Survey became publicly available. we have updated the relevant sections of the manuscript, figures, and tables to incorporate these latest data (now covering 1999–2023 for the Patient Survey and 2014–2023 for NDB Open Data). The main findings and conclusions remain unchanged, but the Results section and Figures 1 and 3 have been revised to reflect the expanded dataset. All such changes are highlighted in the revised manuscript.

RESPONSES TO JOURNAL REQUIREMENTS:

1. Manuscript style requirements:

> Please ensure that your manuscript meets PLOS ONE's style requirements, including those for file naming.

We have carefully revised the manuscript to ensure it meets PLOS ONE's style requirements. Specifically, we have:

- Formatted the main body according to the journal template

- Ensured proper formatting of titles, authors, and affiliations

- Verified all file naming conventions follow PLOS ONE guidelines

2. Funding Statement:

> Please provide an amended statement that declares *all* the funding or sources of support received for the research submitted to the journal.

We have revised our Funding Statement as follows:

"This research was supported by Japan Society for the Promotion of Science (JSPS: https://www.jsps.go.jp/english/) Grant-in-Aid for Scientific Research (C) (Grant Number 23K01927 to MN) and Ministry of Health, Labour and Welfare of Japan (MHLW: https://www.mhlw.go.jp/stf/english/index.html) Health Labor Science Research Grant (Grant Number 23FB1002 to KN). There was no additional external funding received for this study."

3. Role of Funders:

> Please state what role the funders took in the study.

We have added the following statement:

We included the statement above (Funding Statement and Role of Funders) in the cover letter.

4. Data Availability Statement reads:

> Unfortunately, the repository you have noted in your Data Availability statement does not qualify as an acceptable data repository according to PLOS's standards.

We appreciate this guidance. We have now uploaded our compiled dataset to figshare with the following DOI: 10.6084/m9.figshare.29148170. The dataset includes the processed data from all three government sources used in our analysis: the Patient Survey, Comprehensive Survey of Living Conditions, and the NDB Open Data. We have also compiled a detailed list of direct links to the original datasets (Patient Survey, Comprehensive Survey of Living Conditions, and NDB Open Data) as requested by the editorial office. These URLs are now included in the Data Availability Statement and in the supplementary materials.

Our revised Data Availability Statement reads:

"All data underlying the findings described in our manuscript are available from figshare (DOI: 10.6084/m9.figshare.29148170). The original raw datasets analyzed during the current study are available from e-Stat (https://www.e-stat.go.jp/en) for the Patient Survey and the Comprehensive Survey of Living Conditions, and from the Ministry of Health, Labour and Welfare (https://www.mhlw.go.jp/stf/seisakunitsuite/bunya/0000177182.html) for the NDB Open Data.

The original datasets analyzed in this study are publicly available from the following sources:

[list of URLs]"

5. Ethics Statement placement:

> Your ethics statement should only appear in the Methods section of your manuscript.

We have removed the ethics statement from other sections and ensured it appears only in the Methods section of the manuscript.

RESPONSES TO ACADEMIC EDITOR’S COMMENTS:

Comment #1:

> One of the most prominent concerns arises from the divergence between trends in self-reported symptoms and outpatient visits... This incongruity raises important questions.

Thank you for highlighting the important issue of the divergence between trends in self-reported menstrual symptoms and outpatient visits for menstrual disorders. We agree that this paradox is central to interpreting our findings. In the revised manuscript, we have expanded the Discussion section to address this point in greater depth.

Specifically, we now discuss that the observed increase in outpatient follow-up visits for menstrual disorders is unlikely to reflect a true rise in disease burden, as the prevalence of self-reported symptoms has remained stable or declined. Instead, we suggest that several social and systemic factors may have contributed to increased healthcare utilization. These include greater public awareness of menstrual health, reduced stigma surrounding gynecological consultation, improved access to medical care, and the expansion of insurance coverage for LEPs. Such factors may have encouraged more women to seek medical attention for symptoms that were previously self-managed or considered a normal aspect of menstruation. We have also cited recent literature on the impact of awareness, stigma, and health system changes on care-seeking behavior (e.g., Tanaka et al., 2013; Ohde et al., 2008).

The following paragraph has been added to the Discussion section:

“The divergence between the increasing rate of medical follow-ups and the stagnation or slight decline in self-reported symptoms may reflect changes in health-seeking behavior, increased public awareness, reduced stigma, and improved access to care. [15–18, 25, 26]”

Comment #2:

> The results indicate age-specific peaks in outpatient visits and symptom reporting, with notable shifts over time... However, the study does not offer a thorough explanation or control for external variables...

We acknowledge this limitation and have conducted additional age-standardized analysis to address this concern. We have added a new paragraph in the Methods, Results and the Discussion section:

Methods section:

“To account for changes in the age distribution of the female population over time, age-standardized rates were calculated for both the estimated proportion of outpatients with menstrual disorders and endometriosis (from the Patient Survey) and the proportion of persons reporting irregular menstruation or menstrual pain (from the Comprehensive Survey of Living Conditions). Age-specific rates were calculated for each 5-year age group, and the 2015 model female population in Japan was used as the standard population. The age-standardized rate for each survey year was obtained by weighting the age-specific rates by the corresponding age group proportions in the standard population and summing these weighted rates [24].”

Results section:

"After adjusting for changes in the age distribution of the female population, the age-standardized trends in the estimated proportion of outpatients with menstrual disorders and endometriosis were similar to the crude trends (see additional lines in Figure 1A–C). The increase in follow-up visits for menstrual disorders and endometriosis since 2014 remained evident after age standardization."

Discussion section:

“Importantly, age-standardized analyses showed that these trends remained evident even after adjusting for changes in the age distribution of the female population (see Figure 1A–C). This suggests that the recent increase in follow-up visits for menstrual disorders and endometriosis cannot be explained solely by demographic shifts, but likely reflects other factors such as changes in health-seeking behavior, insurance coverage, or public awareness.”

Comment 3:

> The observed 3.8-fold increase in LEP prescriptions from 2014 to 2021, particularly among younger women, is attributable to enhanced accessibility following expanded insurance coverage. Nonetheless, the results focus primarily on prescription volume and drug pricing, without addressing treatment adherence, efficacy, or patient-reported outcomes.

Thank you for this insightful comment. While our study design using aggregate government data cannot directly assess patient-reported outcomes, we acknowledge this limitation and have expanded our discussion to address these important considerations:

“Furthermore, because our analyses were based on aggregated data from government statistics, we were unable to assess individual-level treatment adherence, long-term persistence, symptom improvement, or patient-reported outcomes such as quality of life. Therefore, while we observed a substantial increase in LEP prescriptions and a reduction in economic burden due to generic drugs, we cannot directly determine whether these changes translated into improved clinical outcomes or long-term adherence. Future studies using individual-level data, such as health insurance claims or electronic medical records, are needed to clarify the real-world effectiveness of LEP treatment for menstrual disorders and the impact of generic drug introduction on adherence.”

Comment #3.1:

> Moreover, while drug utilization data are detailed, the lack of correlation with outpatient visit or symptom trends weakens the integrated understanding of treatment effectiveness.

Thank you for your comment regarding the relationship between LEP prescription volume and outpatient visit rates. We agree that these two indicators are derived from different data sources and represent different denominators and populations. Therefore, we believe that formal correlation analysis between these variables would not be statistically meaningful or scientifically appropriate. Instead, we have described the temporal trends of each indicator separately and highlighted their coinciding increases in the Results and Discussion sections. We hope this approach clarifies our intention to provide a descriptive overview rather than to imply a direct statistical association.

Comment #4:

> The discussion and conclusion sections present a well-structured narrative linking the increased use of LEPs to policy changes and access to affordable treatment. However, a deeper academic analysis raises important questions that were not sufficiently explored.

Thank you for this thoughtful observation. We have substantially expanded our Discussion section to include a more critical interrogation of potential contributing factors beyond insurance coverage:

“In addition to the expansion of insurance coverage for LEPs, several other factors may have contributed to the observed increase in follow-up visits for menstrual disorders. Recent studies have highlighted the importance of social and cultural factors, such as increased public awareness, reduced stigma surrounding menstruation, and evolving attitudes toward women's health, in influencing healthcare-seeking behavior among women [25, 33, 34]. In Japan, healthcare avoidance for menstrual symptoms has been associated with attitudes of endurance, limited health literacy, and lack of awareness about available treatment options [25]. Improvements in diagnostic capabilities and greater emphasis on menstrual health in public discourse may also have encouraged more women to seek medical care, even for symptoms previously considered normal. These findings are consistent with international literature showing that social context, education, and economic factors can significantly affect access to and utilization of menstrual health services [33, 34]. Therefore, while the increased use of LEPs and insurance coverage are important, these broader social and systemic changes should also be considered when interpreting recent trends.”

Comment #4.1:

> Moreover, the authors note a discrepancy between increasing medical follow-ups and the stagnation or slight decline in self-reported symptoms; however, this paradox is only superficially addressed.

Thank you for your comment regarding the paradox between increasing medical follow-ups and the stagnation or decline in self-reported symptoms. As noted in our response to Comment #1, we have expanded the Discussion to address this inconsistency in detail.

We now discuss that this divergence may reflect a combination of increased recognition of menstrual disorders as legitimate medical conditions, reduced stigma, and improved access to care, rather than a true increase in symptom prevalence. We also note that socioeconomic and educational factors may influence who seeks treatment, and that further qualitative research is needed to clarify these mechanisms. Relevant literature has been cited to support these interpretations (e.g., Tanaka et al., 2013; Ohde et al., 2008; Getahun SB et al., 2023).

The following paragraph has been added to the Discussion section:

“The divergence between the increasing rate of medical follow-ups and the stagnation or slight decline in self-reported symptoms may reflect changes in health-seeking behavior, increased public awareness, reduced stigma, and improved access to care. [15–18, 25, 26]”

Comment #5:

> Minor revisions: How can future studies address the inability to distinguish between primary and secondary dysmenorrhea in aggregated data?

Thank you for your insightful comment regarding the inability to distinguish between primary and secondary dysmenorrhea in aggregated data. We agree that this is an important limitation. In the Japanese Patient Survey, cases of dysmenorrhea due to endometriosis or adenomyosis that are diagnosed during follow-up are reclassified under "endometriosis" or other relevant categories, and thus are excluded from the "menstrual disorders" category in subsequent surveys. Therefore, among follow-up outpatients, a certain proportion of secondary dysmenorrhea may be excluded. However, it is possible that undiagnosed cases of secondary dysmenorrhea remain within the "menstrual disorders" category, especially at earlier stages of care. As a result, complete differentiation between primary and secondary dysmenorrhea is not feasible using aggregated survey data. Future studies using individual-level data, such as health insurance claims or electronic medical records, would enable more precise classification.

The revised Discussion includes the following paragraph:

“Finally, because the aggregated data used in this study do not allow for the distinction between primary and secondary dysmenorrhea, it is possible that some cases of secondary dysmenorrhea, such as undiagnosed endometriosis or adenomyosis, may remain included in the "menstrual disorders" category, while diagnosed cases are reclassified under "endometriosis" or other relevant categories. Therefore, a certain proportion of secondary dysmenorrhea may be excluded from the follow-up group, but complete differentiation is not possible. Future research using individual-level data is needed to address this issue.”

Comment #5-1:

> What are the underlying reasons why many women still opt for over-the-counter (OTC) medications instead of seeking medical treatment for menstrual disorders?

Thank you for your insightful question regarding the underlying reasons why many women still opt for over-the-counter (OTC) medications instead of seeking medical treatment for menstrual disorders.

Multiple factors contribute to this phenomenon, as supported by the literature and our own findings. First, OTC medications such as NSAIDs are easily accessible, affordable, and familiar, making them a convenient first-line option for many women, especially those with mild or moderate symptoms. Second, social and cultural factors—including the perception that menstrual pain is a normal experience, stigma surrounding gynecological visits, and reluctance to discuss menstruation—can discourage women from seeking professional care. Third, practical barriers such as time

---

## [Editor Report · Decision Letter 1]

Trends in the estimated proportion of outpatients with menstrual disorders and the number of prescribed low-dose estrogen/progestin drugs in Japan: A descriptive study

PONE-D-24-41343R1

Dear Dr. Nakao,

We’re pleased to inform you that your manuscript has been judged scientifically suitable for publication and will be formally accepted for publication once it meets all outstanding technical requirements.

Kind regards,

Marcio Alexandre H. Rodrigues, Ph.D

Academic Editor

PLOS ONE

Additional Editor Comments (optional):

Dear authors,

The changes made following the peer reviewers' comments demonstrate a significant improvement in the manuscript and effectively address the limitations of the study by clarifying its descriptive nature and emphasizing the need for further detailed analyses using individual-level data. The revisions provide a more balanced interpretation of the findings, avoiding overstatements and ensuring transparency regarding the study's scope and methodology. Additionally, the expanded discussion on temporal trends, healthcare utilization, and the impact of social and systemic factors adds depth and relevance to the research. These enhancements have strengthened the overall quality and clarity of the manuscript.

Best Regards

Prof Marcio Alexandre H Rodrigues

Academic Editor
---

## [Editor Report · Acceptance letter]

PONE-D-24-41343R1

PLOS ONE

Dear Dr. Nakao,

I'm pleased to inform you that your manuscript has been deemed suitable for publication in PLOS ONE. Congratulations! Your manuscript is now being handed over to our production team.

Kind regards,

on behalf of

Professor Marcio Alexandre H. Rodrigues

Academic Editor

PLOS ONE